# Microbial Fertilizers: A Study on the Current Scenario of Brazilian Inoculants and Future Perspectives

**DOI:** 10.3390/plants13162246

**Published:** 2024-08-13

**Authors:** Matheus F. L. Andreata, Leandro Afonso, Erika T. G. Niekawa, Julio M. Salomão, Kawany Roque Basso, Maria Clara D. Silva, Leonardo Cruz Alves, Stefani F. Alarcon, Maria Eugenia A. Parra, Kathlen Giovana Grzegorczyk, Andreas Lazaros Chryssafidis, Galdino Andrade

**Affiliations:** 1Microbial Ecology Laboratory, Department of Microbiology, State University of Londrina, Londrina 86057-970, Brazil; m.andreata96@gmail.com (M.F.L.A.); lafonso.bio@gmail.com (L.A.); erikatyemi@gmail.com (E.T.G.N.); julio.salomao@hotmail.com (J.M.S.); kawroquebasso.555@uel.br (K.R.B.); maria.clara.davis@uel.br (M.C.D.S.); leonardo.cruz.alves@uel.br (L.C.A.); stefanialarconf@gmail.com (S.F.A.); maria.eugenia.alcantara@uel.br (M.E.A.P.); kathlen.giovana@gmail.com (K.G.G.); 2Agroveterinary Sciences Center, Santa Catarina State University, Lages 88.520-000, Brazil; andreas.107@gmail.com

**Keywords:** beneficial microorganisms, co-inoculation, soil microbiome

## Abstract

The increasing need for sustainable agricultural practices, combined with the demand for enhanced crop productivity, has led to a growing interest in utilizing microorganisms for biocontrol of diseases and pests, as well as for growth promotion. In Brazilian agriculture, the use of plant growth-promoting rhizobacteria (PGPR) and plant growth-promoting fungi (PGPF) has become increasingly prevalent, with a corresponding rise in the number of registered microbial inoculants each year. PGPR and PGPF occupy diverse niches within the rhizosphere, playing a crucial role in soil nutrient cycling and influencing a wide range of plant physiological processes. This review examines the primary mechanisms employed by these microbial agents to promote growth, as well as the strategy of co-inoculation to enhance product efficacy. Furthermore, we provide a comprehensive analysis of the microbial inoculants currently available in Brazil, detailing the microorganisms accessible for major crops, and discuss the market’s prospects for the research and development of novel products in light of current challenges faced in the coming years.

## 1. Introduction

The growing demand for sustainable agricultural practices with reduced environmental impact has led to a paradigm shift in the use of chemical fertilizers, previously considered essential for maintaining productivity. We now recognize the detrimental effects of these inputs on soil health, the environment, and human well-being. Furthermore, these fertilizers are costly and inefficient, with a significant portion of nutrients becoming immobilized in the soil, rendering them unavailable for plant uptake [1].

The use of growth-promoting microorganisms is steadily increasing, representing a more sustainable practice that can optimize soil nutrient utilization and reduce reliance on chemical fertilizers. The interaction between plants and microorganisms is fundamental for plant development. Microorganisms influence the chemical composition of the rhizospheric environment, impacting root growth, morphology, and permeability through the release of metabolites [2]. Plant growth-promoting rhizobacteria (PGPR) and plant growth-promoting fungi (PGPF) regulate various processes, including nutrient mineralization and solubilization, the production of growth-regulating phytohormones, induction of systemic resistance, enhanced tolerance to abiotic stresses, inhibition of plant diseases through antibiotic production, reduction in ethylene in roots, production of siderophores, and iron chelation [3].

Brazil is one of the World’s largest grain producers, with the agribusiness sector serving as a pillar of its economy, accounting for approximately 23.8% of Gross Domestic Product (GDP) in 2023 [4]. With the growing consumer demand for more sustainable production and products, the use of growth-promoting microorganisms has become a trend in the country, as it has globally. Additionally, the application of inoculants offers a strategy to reduce external dependence on chemical fertilizers. Due to domestic production not meeting internal demand, Brazil is a major importer of fertilizers, accounting for about 80% of the total applied, making it vulnerable to market price fluctuations [5].

According to the National Association of Inoculant Producers and Importers, 2022 saw the delivery of 134.9 million doses by associated companies, representing a notable increase of 30 million doses compared to 2021. There is a growing interest in research and development focused on novel inoculants with more effective, multifunctional strains and formulations that extend the shelf life of these products. Co-inoculation of microbial agents with diverse mechanisms of action and synergistic interactions is a prominent strategy, with a corresponding rise in product registrations in recent years. This practice was employed in 35% of the soybeans cultivated in the country during the 2022/2023 harvest [6].

In this context, this article aims to review the primary mechanisms utilized by PGPR and PGPF to promote growth (Figure 1), highlighting the co-inoculation of strains as a strategy to enhance the efficacy of biological product applications in agriculture. Based on this review, we analyzed the records of microbial inoculants in Brazil, identifying the principal microorganisms and target crops, and discussing market trends in light of current agricultural challenges.

## 2. Plant Growth-Promoting Rhizobacteria (PGPR)

The beneficial interactions between plants and PGPR have emerged as a promising avenue of research for advancing sustainable agriculture and achieving more efficient cultivation practices. These bacteria can exert positive influences on plant development through a variety of mechanisms (Table 1), which will be explored in greater detail in the following sections.

### 2.1. Biofertilization: A Sustainable Approach to Enhance Soil Fertility

Nitrogen is a crucial limiting factor for plant growth, and its availability in the soil directly impacts agricultural productivity. Biological nitrogen fixation (BNF) has emerged as a viable strategy for reducing dependence on nitrogen-based fertilizers. BNF is a complex biological process wherein certain bacteria convert atmospheric gaseous nitrogen into nitrogenous compounds that plants can readily assimilate. This process not only supplies nitrogen to plants but also enriches the soil with essential nutrients, enhancing its fertility and structure [19]. Several genera of rhizobacteria, including *Rhizobium*, *Bradyrhizobium*, *Azospirillum*, and *Azotobacter*, are capable of fixing nitrogen and are frequently utilized as biofertilizers [20].

*Rhizobium* and *Bradyrhizobium* establish nodules on the roots of leguminous plants, providing the specific environment necessary for biological nitrogen fixation. These bacteria possess a unique enzyme called nitrogenase, which can cleave the triple bond of gaseous nitrogen (N_2_) and convert it into ammonia (NH_3_), a form readily usable by plants. The ammonia is then incorporated into essential organic molecules, such as amino acids and proteins, within the plant [21].

In contrast, *Azospirillum* and *Azotobacter* are free-living bacteria that also possess the nitrogenase enzyme, but they do not form nodules on plant roots. The conversion of N_2_ molecules into NH_3_ occurs within the bacterial cells themselves. The ammonia produced in this reaction can be released into the surrounding soil, thereby increasing its overall fertility [22]. BNF by rhizobacteria significantly improves soil quality, primarily by providing a continuous source of nitrogen [23].

Phosphorus (P) is another crucial element for plant growth and development. However, when phosphate-based fertilizers are applied to the soil, they are not always fully utilized by plants. Excess phosphorus often reacts with surrounding metallic cations, such as calcium (Ca^2+^), iron (Fe^3+^), and aluminum (Al^3+^), forming insoluble phosphate compounds. These phosphates (calcium phosphate, iron phosphate, and aluminum phosphate) exhibit low solubility in water, restricting the availability of P to plants [24].

*Bacillus*, *Pseudomonas*, *Acinetobacter*, and *Pantoea* species play a crucial role in enhancing P availability in the soil, particularly in its insoluble forms. These bacteria, known as PSB (phosphate-solubilizing bacteria), produce and release various organic acids (formic, acetic, propionic, lactic, and succinic acids). The hydroxyl and carboxyl groups of these released acids react with phosphate minerals, effectively dissolving and converting them into forms that plants can readily uptake [25]. Through the production of these organic acids, these bacteria also improve the availability of other essential nutrients, such as potassium (K), a critical nutrient for plant growth. Potassium is often applied in soluble forms but tends to become rapidly fixed in the soil [26].

The combined use of nitrogen-fixing bacteria with phosphorus and potassium-solubilizing bacteria offers a viable alternative for reducing the application of chemical inputs in agricultural fertilization, contributing to a more sustainable and economically viable approach to agriculture [27].

### 2.2. Protection against Oxidative Stress in Adverse Environmental Conditions

Reactive oxygen species (ROS), such as the superoxide radical (O_2_^−^), hydrogen peroxide (H_2_O_2_), and hydroxyl radical (OH^−^), are highly reactive molecules produced in plants as byproducts of oxygen metabolism, especially during photosynthesis and respiration. In stressful situations, such as drought, salinity, extreme temperatures, pathogen infection, and exposure to pollutants, ROS production increases significantly. Although at low levels they act as signaling molecules, at high concentrations they cause oxidative damage to proteins, lipids, and DNA, resulting in cellular dysfunction and, in severe cases, cell death [28].

Antioxidants play a vital role in protecting plants from oxidative stress and are classified as either enzymatic or non-enzymatic. Enzymatic antioxidants, including superoxide dismutase (SOD), catalase (CAT), peroxidases (POD), glutathione reductase (GR), and ascorbate peroxidase (APX), neutralize ROS through specific biochemical reactions and are produced by both plants and certain PGPRs [29]. Non-enzymatic antioxidants, such as ascorbic acid (vitamin C), glutathione (GSH), carotenoids, tocopherols (vitamin E), and flavonoids, directly scavenge ROS. While these compounds are primarily synthesized by plants, some PGPRs are also capable of producing ascorbic acid and glutathione [30].

Several species of rhizobacteria, including *Bacillus pumilus*, *Bacillus subtilis*, and *Pseudomonas* sp., contribute to the protection of plants from oxidative stress. They achieve this by either directly producing antioxidants or by stimulating plants to increase their own antioxidant production [31]. This collective action of antioxidants creates a protective network that shields plant cells from oxidative damage, ensuring the maintenance of plant health and growth even in adverse environmental conditions [32].

### 2.3. Production of Phytohormones by Rhizobacteria

Phytohormones are naturally occurring chemical substances within plants that regulate a broad spectrum of physiological and developmental processes. Collectively, these hormones can interact synergistically or antagonistically, governing various aspects of plant growth, development, and adaptation to the environment. The principal phytohormones include auxins, cytokinins, gibberellins, abscisic acid, ethylene, salicylic acid, and jasmonates. In addition to being synthesized by plants themselves, the production of many of these hormones can be induced or directly produced by PGPRs such as *Pseudomonas*, *Azospirillum*, *Bacillus*, *Burkholderia*, *Serratia*, and others [33].

The main growth-promoting mechanism employed by rhizobacteria is the production of auxin, particularly indole-3-acetic acid (IAA). Over 80% of bacteria associated with the rhizosphere are capable of synthesizing IAA [34]. Auxins modulate growth processes in both the aerial and root portions of plants. The increase in root size, induced by bacterial IAA, enhances the surface area available for nutrient and water uptake, thereby improving overall plant vigor [35]. Additionally, research in *Arabidopsis thaliana* has shown that auxins improve resistance to water stress by positively regulating the activity of antioxidant enzymes, aiding in the mitigation of the effects of reactive oxygen species generated under stressful conditions [36].

Cytokinins (CKs) are known to regulate a wide array of plant growth, development, and physiological traits, including seed germination, apical dominance, flower and fruit development, and leaf senescence. CKs also participate in a diverse range of responses to biotic and abiotic stresses. Although the precise mechanisms of action are not fully understood, numerous studies have observed an increased endogenous concentration of CKs in plants under stress or altered stress responses upon the addition of exogenous CKs [37]. Bacterially produced CKs can induce resistance in plants against bacterial and fungal pathogens [38]. However, they can also stimulate plant responses to herbivore attacks by promoting the expression of wound-inducible genes, leading to the accumulation of insecticidal compounds [39].

Gibberellins (GAs) are phytohormones that promote various aspects of plant growth, including stem elongation, seed germination, flowering, and fruit development. They are also crucial for breaking seed dormancy [40]. In contrast, abscisic acid (ABA) is associated with stress responses and dormancy. It inhibits growth, promotes stomatal closure to minimize water loss, and induces seed dormancy, aiding plants in surviving adverse conditions such as drought or salinity [41]. At the molecular level, GAs and ABA exert antagonistic regulation on the expression of specific genes. GAs activate genes involved in cell growth, while ABA induces genes related to stress response protein synthesis. The signaling pathways of these hormones involve specific receptors and transduction mechanisms that modulate the activity of transcription factors, leading to appropriate cellular responses. The interaction between gibberellins and ABA exemplifies the antagonistic regulation of critical processes within the plant life cycle. The balance between these hormones enables plants to adjust their growth and development in response to environmental fluctuations, ensuring survival and adaptation to changing conditions [42].

Some PGPRs play a vital role in plant health by producing and modulating plant hormones such as salicylic acid (SA), jasmonate (JA), and ethylene (ET), which are essential for plant responses to biotic and abiotic stresses. Salicylic acid (SA), produced by PGPRs like *Pseudomonas* and *Bacillus*, induces systemic acquired resistance (SAR), enhancing defenses against pathogens [43,44]. Jasmonate (JA) and ethylene (ET), influenced by PGPRs, trigger induced systemic resistance (ISR) and increase defense against herbivores. The coordinated interaction among SA, JA, and ET, mediated by PGPRs, offers significant benefits to plants. Cross-talk between these signaling pathways allows the plant to fine-tune its responses to multiple stressors, promoting both defense mechanisms and growth. While SA is most effective against biotrophic pathogens, JA and ET play crucial roles in defense against herbivores and under abiotic stress conditions. The combined effect of these responses provides a robust defense strategy against a wide range of stresses, increasing plant resilience [45,46].

The capacity of PGPRs to directly induce or produce phytohormones underscores their crucial role in promoting plant growth and enhancing stress resilience, providing promising avenues for increasing agricultural productivity.

### 2.4. Production of 1-Aminocyclopropane-1-Carboxylic Acid (ACC) Deaminase

Ethylene is a gaseous hormone produced by plants and plays a vital role in their development and adaptation to adverse conditions. During biotic stresses, such as pathogen attacks, the increase in ethylene levels triggers a series of defense responses, including the production of phytoalexins and the activation of resistance genes [47]. Similarly, in abiotic stresses such as drought and salinity, ethylene assists plants in survival by adjusting growth, closing stomata to reduce water loss, and promoting the synthesis of antioxidant enzymes to combat oxidative stress. However, when ethylene levels exceed what is needed, undesirable effects can occur, such as the inhibition of root and stem growth, reduction in photosynthetic rate, and even induction of premature senescence. Additionally, excess ethylene can lead to premature leaf and flower drop, compromising the plant’s ability to photosynthesize and reproduce. Therefore, the precise regulation of ethylene levels is crucial to ensure the optimal growth and development of plants [48].

Some PGPR produce ACC deaminase, an enzyme that reduces ethylene concentrations in the plant by cleaving the precursor ACC into ammonia and α-ketobutyrate. This mechanism helps reduce damage caused by various stresses, particularly environmental ones, such as in saline soils [49], heavy metals [50], and water stress [51].

### 2.5. Chemical Signals: Volatile Organic Compounds (VOCs)

Volatile organic compounds (VOCs) are small chemical substances (<300 Da) widely released by most living organisms. Bacterial VOCs encompass a variety of categories, including alcohols, benzenoids, aldehydes, alkenes, acids, esters, terpenoids, and ketones. The ones produced by rhizobacteria can trigger several intra- and interspecies responses; in plants, we highlight the plant growth promotion and induction of systemic resistance effects [52].

Once liberated, these compounds become powerful chemical signals displaying roles in the modulation of microbial community growth, movement, and virulence; the expression of genes related to the management of stress; secondary metabolites production; and quorum sensing. Plant responses to VOCs include growth (shoot and roots, especially), enhanced nutrient uptake, and induced systemic resistance. Animal responses to microbial VOCs include the attraction of beneficial insects and repulsion of harmful pests [52,53].

Among the genera of rhizobacteria identified as VOC producers are species of *Bacillus*, *Serratia*, *Enterobacter*, and *Pseudomonas* [54]. On the suppressive effects of bacterial VOCs, for example, molecules produced by *Pseudomonas* are reported to impair *Ralstonia solanacearum* growth and virulence [55], suppressive effects were also reported over *Rhizoctonia solani*, *Botrytis cinerea*, and *Phytophthora cinnamomi*, impairing the different process of the phytopathogenic fungi [56]. Examples of beneficial effects from VOCs produced by fungi are presented in Table 2.

The multiple categories of compounds and the intricate biochemical pathways involved in their signaling make the complete elucidation of these mechanisms a challenge. However, studies indicate that microbial VOCs are strongly associated with promoting plant growth, possibly by modulating the synthesis and/or metabolism of phytohormones produced by rhizobacteria or by the plants themselves [33].

### 2.6. Production of Siderophores

Siderophores are secreted molecules that can be used both as biofertilizers and/or biocontrollers. As biofertilizers, they optimize the availability of iron in the root zone, and at the same time, their biocontroller activity relies on competing with phytopathogens for access to iron [67].

From a structural perspective, a typical siderophore presents one or more well-known functional groups such as hydroxamate, catecholate, or carboxylate. These functional groups establish bonds with iron ions, forming stable complexes. Although siderophores are widely recognized for their affinity with iron, it is important to emphasize that some of these organic molecules can also form complexes with other metals, including zinc, copper, manganese, and others [68].

There are two theories regarding how plants absorb iron from microbial siderophores. According to the first theory, siderophores with high redox potential donate iron (Fe^2+^) to the plant’s transport system after being reduced. This happens when siderophores with Fe^3+^ are transported to the roots, where reduction occurs in the apoplast, retaining Fe^2+^ and increasing iron concentrations in the roots. Following the second theory, microbial siderophores can bind to iron in the soil and exchange ligands with plant phytosiderophores, a process dependent on various factors such as complex stability, concentrations, and root environment conditions [69].

In addition, the high affinity of siderophores for iron results in limiting the access of this element by other microorganisms, suppressing the population of phytopathogenic agents in the rhizosphere. The reduction in disease incidence strengthens plant health and reduces the need for high doses of chemical control agents [70].

## 3. Plant Growth-Promoting Fungi (PGPF)

Some genera of saprophytic fungi present in the rhizosphere are capable of promoting growth by colonizing the plant’s roots, primarily through increased nutrient absorption, phytohormone production, and the induction of systemic resistance. Among the most common are *Aspergillus*, *Fusarium*, *Penicillium*, *Piriformospora*, *Phoma*, and *Trichoderma* [71]. Here, we focus on fungi from *Trichoderma* genus and arbuscular mycorrhizal fungi (AMF). The latter are obligatory symbionts of plant roots and have a notable effect on nutrient acquisition and resistance to abiotic stress [72] (Table 2).

### 3.1. Trichoderma spp. as Biocontrol Agents and Fertilizers

Fungi of the genus *Trichoderma* have remarkable abilities in the biological control of pathogenic fungal agents that harm plant growth. This activity in controlling fungi is well documented in the literature, and various mechanisms of action are described [73].

This effect can occur through direct or indirect mechanisms. Direct mechanisms rely on competition for resources, such as space and nutrients. *Trichoderma* spp. colonize surfaces and substrates, occupying spaces and consuming nutrients.

It limits the growth of pathogens, restricting their establishment in plants. Chitinases and Beta-1,3-glucanases produced by *Trichoderma* spp. are vital keys to its action against other fungi. The production of these enzymes is part of the process known as mycoparasitism, where, upon recognizing chemical signals from the cell wall of the pathogen, *Trichoderma* hyphae grow towards and wrap around the hyphae of the pathogenic fungus. They then initiate the secretion of enzymes that degrade and create pores in the cell wall, through which *Trichoderma* obtains nutrients [74,75].

Regarding indirect mechanisms, the activation of the plant’s defense system is observed. This is because *Trichoderma* spp. emit chemical signals that stimulate the production of defensive compounds in plants, such as phytoalexins and enzymes [76,77].

Organic compounds, such as VOCs, are also produced by *Trichoderma* spp. and influence the germination of fungal spores, inhibiting their development. VOCs produced by *Trichoderma koningiopsis* T-51 inhibited the mycelial growth of *B. cinerea* by 73.78% and of *Fusarium oxysporum* by 43.68%. Additionally, there was a reduction in conidial germination and a delay in germ tube elongation [59].

It is important to highlight that the production of metabolites can vary between different species and strains of *Trichoderma*, as well as the environmental conditions. These metabolites play critical roles in the ecology of *Trichoderma* and their interactions with plants and other microorganisms, consolidating their role in controlling plant diseases and promoting soil health. Additionally, these fungi also play a vital role in improving nutrient and water uptake [78,79,80].

Out of 251 *Trichoderma* isolates from the Amazon rainforest soil, 49 demonstrated phosphate solubilization capacity. The production of organic acids, such as lactic acid, fumaric acid, ascorbic acid, malic acid, gluconic acid, D-isocitric acid, phytic acid, and citric acid, was observed. Additionally, two strains showed growth-promoting activity in soybean plants in a greenhouse setting [57]. High solubilization activity and growth promotion were also confirmed for *Trichoderma asperellum* UFT 201 in soybean cultivations in a greenhouse, and these results were later replicated in a field experiment [81,82].

The combination of these action mechanisms confers significant benefits to the plant. After selecting four strains with positive inhibitory activity against *F. oxysporum*, phosphate solubilization, and IAA production, an evaluation of tomato seed inoculation confirmed an increase in chlorophyll levels, aerial part length, fresh and dry weight of both the aerial part and roots, and a reduction in wilt disease caused by *F. oxysporum* ranging from 10 to 30% [83].

### 3.2. Arbuscular Mycorrhizal Fungi: Extensions of Roots in Soil

AMF have a symbiotic relationship with plant roots. Intracellular hyphae form arbuscules, specialized structures that penetrate the root epidermis to exchange nutrients with hosts [84]. AMF provide many benefits to plants, such as increased photosynthetic rates, improved soil quality, influence on atmospheric CO_2_ fixation [85], enhanced nutrient absorption (particularly P), increased water uptake, protection against pathogens, and assistance in obtaining micro- and macronutrients [86].

Benefits arise from AMF functioning as extensions of roots, penetrating other parts of the rhizosphere. Additionally, their hyphae are thinner compared to roots, allowing them to access a greater soil volume. For protection against water deficit, AMF hyphae enhance access to small soil pockets [87]. Also, the presence of phytohormones like abscisic acid modulates plant mechanisms related to water deficit, such as aquaporins and transpiration [88]. AMF can also increase flood tolerance, improving growth and assisting in phosphorus absorption through osmotic adjustment performed by the fungi [89].

Heat stress is also attenuated by AMF, which increases nutrient and water absorption, along with improved photosynthetic rates. Other effects are the accumulation of proline and sugars, sodium reduction, increased carbon, and homeostasis [90]. These mechanisms, aiding in extreme temperature tolerance, also contribute to salinity tolerance [91]. Therefore, AMF plays a pivotal role in terrestrial ecosystems’ microbiomes, helping host plants and ecosystem maintenance [92].

Among its roles in plant health regarding enhanced nutrient obtention, P absorption is the most thoroughly described and recognized as the primary regulator of the association. In conditions of high available concentrations of this mineral in the soil, the plant tends to reduce its investment in the association, whereas, under conditions of low availability, plant investment is higher. This enhanced P absorption occurs directly, through the absorption and transport of P via the hyphae to the arbuscules, and indirectly, by stimulating phosphate-solubilizing bacteria in the soil [93].

This stimulation occurs in the region known as the mycorrhizosphere, which is the zone around roots colonized by mycorrhizal fungi, directly influenced by hyphal and root exudation [94]. Composed of hyphae with a diameter of approximately 0.2 μm, much finer than roots but significantly denser, the mycorrhizosphere is broader than the rhizosphere and acts as a matrix, facilitating interactions between bacteria and fungi. Extraradical fungal hyphae form the hyphosphere, a region influenced by AMF hyphae, that bring about physical, chemical, and biological alterations. Examples of these alterations include soil particle aggregation and the direct delivery of water to the host plant [95,96].

Fungal hyphae release carbon-rich compounds promoting bacterial growth, and, on the other hand, hyphae can also produce compounds that signal or inhibit other microorganisms’ growth [97], affect soil pH, maintain a liquid film, and serve as bacterial concentration nodes, creating a microbiome that supports soil maintenance [98].

Through the hyphal exudation of signaling molecules and compounds, AMF can influence the surrounding soil, modifying the local microbiota, referred to as the hyphosphere effect. It can also impact weathering, where exudates act as chelators, destabilizing mineral surfaces [99,100]. Thus, the hyphosphere provides an excellent habitat for other microorganisms, supporting the soil microbiome and influencing various soil processes [101].

## 4. Co-Inoculation of Beneficial Microorganisms

In recent years, there has been increased interest in combining microorganisms with similar activities but different mechanisms of action. This management strategy has been primarily studied in the control of diseases caused by nematodes, fungi, and insects, resulting in a reduction in the use of chemical products and a decreased selection of resistant pathogens. [102,103].

The interaction between growth-promoting microorganisms does not always result in the improvement in plant development or the biocontrol of pathogens; it is subject to several factors, such as the genetic variability of native bacteria, hosts, and environmental factors, such as light, temperature, and the organic matter of the soil [104].

These interactions are classified as synergistic, antagonistic, or non-interactive/additive, according to the observed effect. Synergy occurs when a combined effect is observed, meaning that co-inoculation provides a greater effect compared to the application of these agents individually. On the other hand, in antagonism, this effect is negative, whereas in non-interactive interactions, there is no impact compared to the isolated application of each microorganism [105].

An example of a synergistic interaction is the co-inoculation between AMF and PGPR, which has been examined by several studies, especially nitrogen-fixing bacteria (NFB). In general, gains in biomass and minerals were achieved, particularly in nitrogen and phosphorus rates [89,106].

The activity of AMF and NFB in the rhizosphere is essential for plant nutrition, and various factors contribute to a tripartite symbiosis. These microorganisms do not compete for the same colonization sites, indicating coexistence and possibly functional interactions. Furthermore, the inoculation of AMF has already been shown to be important in establishing NFB and improving nodulation, as nodulation is dependent on high levels of P, which can be increased by AMF colonization [104]. It is also worth noting the importance of the correct combination of strains to achieve synergy in the co-inoculation of these microorganisms [107] (Table 3).

## 5. Scenario of Inoculants Registered in Brazil

According to the Brazilian Ministry of Agriculture (MAPA), there are 636 inoculants with 713 registrations (some inoculants are registered for multiple crops) in Brazil (as of April 2024, based on the Ater digital platform provided by the Brazilian Federal Government). These registrations encompass 37 crops, ranging from vegetables like lettuce, cabbage, and potatoes, but primarily focusing on grains [114]. Approximately 50% of these registrations are for soybean cultivation, the most extensively planted crop in the country, covering an area of approximately 45 million hectares [115]. Bean and maize, also among the most cultivated crops in the country, have 73 and 70 registrations, respectively. Jack bean (35), peanut (36), and wheat (23) follow as the crops with the next highest number of inoculant registrations. The remaining registrations are distributed among the other 31 crops, with lettuce notably having eight (Figure 2).

### 5.1. Formulations: Single and Co-inoculation

Among these registrations, 622 consist of formulations with a single microorganism, with a significant portion being NFB such as *Bradyrhizobium* spp., *Azospirillum brasilense*, and *Rhizobium* spp. For soybeans, for example, there are 189 registrations of products with *Bradyrhizobium japonicum*, 57 for *Bradyrhizobium Elkani*, and 52 with the co-inoculation of *B. japonicum* and *B. elkani*. For beans, 65 out of 73 registrations are for *Rhizobium tropici* and 6 are for *A. brasilense*. In maize cultivation, 45 out of 70 registrations contain *A. brasilense* [114].

The introduction of these inoculants, commencing in 1964, played a crucial role in establishing Brazil’s competitiveness in grain production and reducing the need for nitrogen fertilization. Recent years have seen the implementation of formulations co-inoculating two NFB strains, resulting in increased yields compared to individual inoculation. This co-inoculation strategy is expected to gain further traction in the coming years [116]. According to Telles et al. (2023), during the 2019/2020 harvest season, co-inoculation with *Bradyrhizobium* spp. and *A. brasilense* was employed in 25% of all soybean-cultivated areas in Brazil, yielding cost savings of 15.2 billion dollars by replacing urea application with biological nitrogen fixation. Additionally, it generated an estimated profit of 914 million dollars [117]. In the 2022/2023 harvest season, it is estimated that this practice was applied in 35% of soybean production in the country, a 10% increase compared to the previous season [6].

When analyzing a subset of the most important crops (Table 4), in addition to NFB, there are also registrations of plant growth-promoting rhizobacteria (PGPR) from the genus *Bacillus* and *Pseudomonas fluorescens*. In certain cases, co-inoculation between two strains is implemented, such as *B. subtilis* and *B. megaterium* for soybean, maize, and bean crops, *B. megaterium* and *Lysinobacillus* sp for maize, strains that promote growth by providing phosphorus availability, and *B. licheniformis* and *Bacillus aryabhattai* act on increasing stress tolerance in maize. The *B. megaterium* B119 and *B. subtilis* B2084 strains, registered as phosphate solubilizers, are multifunctional microorganisms that produce high levels of phytohormones, fix nitrogen, and produce siderophores. Additionally, they occupy distinct ecological niches, one being a rhizosphere isolate and the other an endophyte of maize roots, thus expanding the spectrum of action and specificity for maize cultivation, with gains in productivity and grain P content in low-fertility soils [118].

In addition to *P. fluorescens* and *A. brasilense* for maize and soybean crops, there are also strains of *B. subtilis* and *B. elkani* for soybean. In these latter two cases, there are the co-inoculation of strains with different activities, biological nitrogen fixation, and phosphorus solubilization. This is an excellent alternative, given that some studies have already shown that a higher P content can influence the amount of N fixed in the soil [104,119]. The addition of K-solubilizing bacteria to these formulations could be a strategy used to increase the absorption efficiency of the three main macronutrients applied in Brazilian soils.

Co-inoculation with three strains is also observed, such as *B. subtilis*, *Bacillus amyloliquefaciens*, and *B. pumilus* for promoting growth in soybean and maize, and *P. fluorescense*, *B. amyloliquefaciens*, and *Priestia megaterium*, as well as *B. subtilis*, *B. elkani*, and *Parabhurkodelia nodosa* for soybean. This last species, introduced in the year 2024 to the market, isolated only in Brazilian soils so far, exhibits nitrogen fixation activity, among other beneficial interactions such as phosphate solubilization, hormone production, siderophore synthesis, and ACC deaminase activity [120].

In terms of fungi, the number of registrations is reduced to five registrations of *Trichoderma* for soybean and five registrations of AMF for soybean and maize. Among these, one formulation contains eight strains (4 *Trichoderma harzianum*, 3 *Trichoderma asperelloides*, and 1 *Trichoderma koningiopsis*), which are registered for soybean, alongside the other *Trichoderma* products. As for AMF, the registrations are for soybean and maize, containing a strain of *Rhizophagus intraradices* or a double inoculation with *R. intraradices* and *Claroideoglomus claroideum* (Table 4).

### 5.2. Biofertilization

Brazil consumes approximately 8% of the global fertilizer market, accounting for about 85% of the fertilizers applied domestically, with an annual expenditure of USD 25 billion. In 2020, domestic production of nitrogen fertilizers met only 4.3% of the national demand, a figure that has been declining due to the accelerated growth of the agricultural sector outpacing fertilizer production. In 2010, this figure was 20.7%. Regarding phosphorus (P) and potassium (K), imports accounted for 73% and 97%, respectively, highlighting a high external dependence that exposes Brazil’s primary economic sector to fluctuations in the international fertilizer market. This situation is exacerbated when considering the utilization efficiency of these inputs, ranging from 50 to 70% for N, 15 to 50% for P, and 50 to 70% for K. This inefficiency is primarily attributed to the inadequate transfer of fertilization technologies designed for temperate climates to tropical regions, which experience higher rainfall, greater microbial activity, and highly weathered soils. In 2022, the Brazilian Federal Government introduced the National Fertilizer Plan (NFP), aimed at reducing the importation of these inputs from 85% to 45% by 2050. The development of more efficient biological inputs is one of the strategies outlined in the plan [4].

Brazil’s inoculants market is experiencing substantial growth. NFB-based inoculants are well established in the production of various crops, most notably maize and soybeans. Nonetheless, the adoption of inoculants with alternative growth-promoting mechanisms remains in its early stages. Analogous to NFB in the context of nitrogen fertilization, AMF and PSB have the potential to revolutionize phosphate fertilization practices in Brazil. More than 70% of phosphorus fertilizers applied in Brazilian soils are imported, generating a variable cost according to the dollar exchange rate, and are poorly utilized. It is believed that, on average, only 30% of this phosphorus is absorbed by plants, with a significant portion of the surplus accumulating in the soil. Brazilian soils contain elevated concentrations of P in insoluble forms, a consequence of decades of fertilization in iron (Fe) and aluminum (Al)-rich soils, culminating in the rapid immobilization of this vital nutrient. It is estimated that by 2018, Brazil accumulated a staggering 33.4 teragrams (Tg) of phosphorus in its soil. This estimation is derived from an analysis of the ratio between phosphorus inputs from organic manure and mineral fertilizers and the phosphorus harvested by crops annually, dating back to 1967, a seminal year that marks the commencement of intensive phosphate fertilizer application. This accumulated phosphorus reservoir possesses an estimated economic value of USD 22 billion [121].

The inoculation of AMF and PSB is an important alternative to enhance the efficiency of P absorption, reducing the need for application and utilizing the soil’s legacy P [121,122]. However, the application of these microorganisms is still not widely disseminated in Brazil. For example, AMF products available are imported and do not utilize native strains from the country, highlighting a gap in investment in technologies for strain research and the development of production systems for these microorganisms. Cely et al. (2016) demonstrated the ability of the arbuscular mycorrhizal fungus (AMF) *R. clarus*, cultivated in vitro in association with transformed carrot roots, to enhance the effectiveness of P application in soybean and reduce the required dose by half in cotton under field conditions. This cultivation method is efficient for producing large quantities of propagules, free from contaminants and with homogenous batches, but still requires scaling up [62].

In the case of PSB, although products are already available on the market, they have not yet achieved widespread adoption. However, the trend indicates an increase in product availability, with the registration of new species and a rise in the number of doses applied in upcoming harvests. Mosela et al. (2022) confirmed the high capacity of the *Bacillus velezensis* strain Ag 75 to improve P application efficiency in soybean and maize crops under field conditions, a species not yet commercially available [15].

Like phosphate fertilizers, potassium fertilizers are largely imported, around 97%, a percentage that has been increasing in recent years due to the accelerated growth of agriculture [115]. It is estimated that the efficiency of applied potassium (K) in the soil is 66%, with a considerable portion remaining stocked up in the soil and 13% lost through erosion and leaching processes [123].

Brazil is a country endowed with extensive mineral resources, among which potassium-containing rocks hold potential as a nutrient source for agriculture. Currently, many rock-derived powder products are registered as remineralizers by MAPA (the Brazilian Ministry of Agriculture). Among these, some are primarily composed of glauconite, a mineral found in rocks such as greensand, slate, and glauconitic siltstone, serving as a significant source of K [124]. However, the availability of potassium from these sources depends on the physicochemical conditions of the soil; generally, only 1 to 2% is directly available for plant uptake, with the remainder existing in non-exchangeable forms in the soil [123]. To enhance potassium availability, investments are needed in research and development of strategies to facilitate the rapid release of these nutrients for plant absorption. These processes can be thermal, chemical, or biological in nature [124,125].

Research based on biological solubilizers is still limited but demonstrates potential and should be further explored, as these methodologies tend to be less costly compared to thermal and chemical processes. In a study utilizing greensand as a K source, *Burkholderia* sp. and *Bacillus* sp. strains were able to extract 71.3% and 53.6% of K, respectively, compared to the control [125]. Matias et al. (2019) demonstrated the ability of the bacterium *Acidithiobacillus thiooxidans* to solubilize K from greensand, primarily through medium acidification, from pH 4.2 to 0.57 after 49 days of incubation [126].

According to the National Fertilizer Program (NFP), by 2050, Brazil intends to be a reference in the significant production of remineralizers and other alternative sources of potassium derived from silicate rocks. Exploring strains with high potassium solubilization activity and developing biological products that enhance the availability of this nutrient represent gaps to be addressed in the Brazilian market [5].

### 5.3. Abiotic Stress Mitigation

Other inoculants that are expected to see increased registrations in the coming years are stress mitigators. The effects of adverse weather conditions are already being felt in Brazilian agriculture. According to the National Supply Company (Conab), the estimate for the harvest in the 2023/2024 season is 8% lower compared to the previous season, a reduction of around 25.7 million tons. This loss of productivity is directly related to the delayed onset of rains in the Midwest, Southeast, and Matopiba regions, high temperatures, irregular and poorly distributed rainfall, and periods of drought lasting more than 20 days. In addition to the impacts on productivity, these effects may, in the long term, affect the cultivable regions for Brazil’s main crops, reducing areas suitable for soybean and maize cultivation [127]. Currently, there are already *Bacillus* sp. strains on the Brazilian market for reducing water stress, registered for soybean and maize crops. The CMAA1363 strain of *B. aryabhattai* was prospected from cactus rhizospheric soil in the Caatinga biome of the Brazilian semiarid region during the dry season [128]. Evaluations in four different edaphoclimatic regions confirmed that this *B. aryabhattai* strain is a potent growth promoter, increasing maize yields by 5.9 to 43.7% [129]. In a study conducted on maize in an agroecological system, inoculation with the CMAA1363 strain partially mitigated the impacts of water and salt stress through morphophysiological characteristics, such as increased leaf area and plant height [130].

However, given the magnitude of the economic impact that these climate changes can cause, there is a trend towards increased investment in research and development of biological products to mitigate the effects of climate change on agriculture and a greater variety of strains and formulations covering more cultures. The impact of AMF inoculation on plants under abiotic stress is well documented in the literature. In addition to their role in nutrient absorption, these microorganisms have high potential as multifunctional inoculants and for co-inoculation with PGPB [131,132].

## 6. Future Perspectives

The application of inoculants has been growing worldwide. With increasing research on the impact of chemical inputs on the environment and human health, and the recognition of the importance of microorganisms in nutrient cycling and maintaining soil quality, production systems are undergoing a transition towards more sustainable agriculture. The use of these inputs is among the strategies used to achieve the sustainable development goals set by the United Nations General Assembly and is seen as essential for maintaining global food security [133]. As one of the largest agricultural producers globally, Brazil has been rapidly adapting its legislation and accelerating the registration process for biological products. With the collaboration of companies, research institutes, and the consumer market, processes and parameters are being defined to ensure product quality, from research for registration to final product quality control. Given the emergence of new technologies, this regulatory process must be continuous in the coming years.

In this context, research on the impact of PGPBs on various crops has been increasing. It is likely that more specific inoculants will be registered, tailored to the needs of each crop. With the aid of precision agriculture, the application of these bio-inputs can be targeted based on the characteristics of the region, soil, and plant, optimizing their utilization.

In the case of Brazil, the application of inoculants will be essential for reducing external dependence on fertilizers and improving the utilization of nutrients retained in the soil. Investment in research on P-solubilizing microorganisms is necessary, as they are still not well established in the market but have great potential for growth in the number of registered products and are expected to have more doses applied in the coming harvests. It is also crucial to encourage research on K-solubilizing bacteria, which currently have limited studies, as they may hold the key to utilizing alternative sources of this nutrient. The application of inoculants will also be a vital tool for mitigating the impact of climate change. To achieve this, research should focus on increasingly efficient strains and more resistant formulations that extend shelf life and maintain the performance of microorganisms in the field, even under adverse environmental conditions.

The utilization of new biotechnologies will be of paramount importance for the development of higher-quality, more complex bio-inputs in a shorter timeframe. Advances in sequencing technology will aid in the identification and prospecting of improved strains. Through mining the genomes of competent strains, we can identify and reveal the functions of genes involved in specific mechanisms. More research is also needed to understand the interaction between microorganisms and plants in the soil, identifying beneficial metabolites and compounds produced by growth-promoting microorganisms. Based on these data, genetic engineering can be employed to optimize the activity of strains, but the application of products with modified microorganisms requires debate regarding their impact on soil ecology, ethical considerations, and regulatory frameworks [134].

Regarding the extension of shelf life, new formulations of agricultural inoculants have evolved significantly, incorporating carriers and encapsulation technologies that enhance the efficacy and stability of beneficial microorganisms. Innovative carriers, such as biodegradable polymers and biomass matrices, provide an ideal environment for the survival of inoculants, ensuring controlled and prolonged release. Encapsulation, in turn, protects microorganisms from adverse conditions like heat and desiccation and enables targeted release in the soil, optimizing root colonization. These innovations not only improve the viability of inoculants but also enhance agronomic benefits, such as increased productivity and crop sustainability [134].

## 7. Conclusions

The utilization of PGPR and PGPF as biological inoculants has proven to be efficient and capable of generating economic benefits for agricultural production systems, as evidenced by the individual or co-inoculated use of *Bradyrhizobium* spp. and *A. brasilense*, contributing to the sustainability of Brazilian agriculture. With the well-established use of NFB in the country, the focus in the coming years should be on establishing and scaling up the application of PSB, along with the registration of new strains and more efficient formulations. The development of AMF-based products is also crucial, as their multifunctionality makes them valuable tools for agriculture, contributing to nutrient availability, mitigating the impact of abiotic stress, and offering synergistic potential in co-inoculation with PGPB. Additionally, investment in research is needed to expand our understanding of the potential of K-solubilizing bacteria in extracting this nutrient from alternative sources, which can help reduce external dependence on chemical fertilizers.

Climate change is expected to be one of the major challenges of the coming decades and could drastically reduce Brazilian agricultural cultivation areas and production. This problem requires constant monitoring, research, and development to mitigate its effects on agriculture. The selection of improved strains and the development of new inoculant formulations with extended shelf life and enhanced field performance will be crucial in addressing the challenges posed by climate change and ensuring sustainable agricultural practices.

## Figures and Tables

**Figure 1 plants-13-02246-f001:**
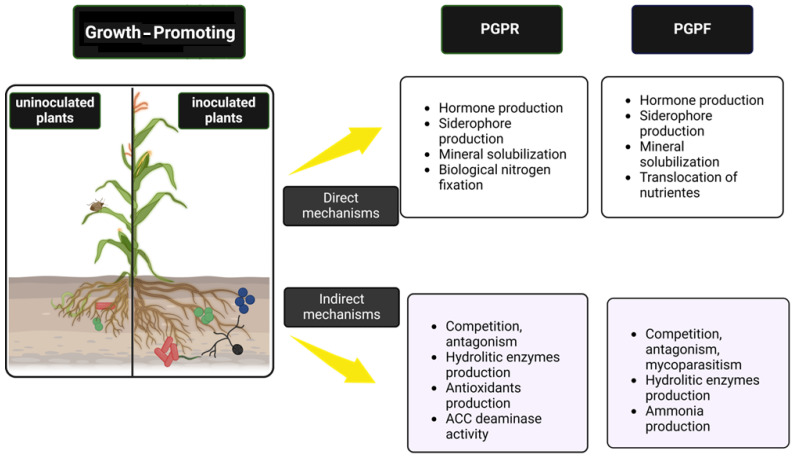
Direct and indirect mechanisms of PGPR and PGPF for plant growth promotion in the soil.

**Figure 2 plants-13-02246-f002:**
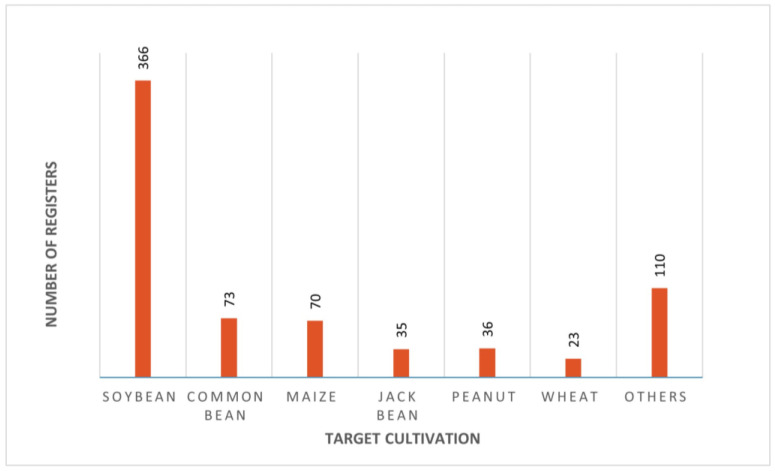
Number of inoculants registered for the main crops in Brazil.

**Table 1 plants-13-02246-t001:** Some examples of PGPR growth promotion mechanisms.

Specie	Mechanism of Growth Promotion	Culture	Reference
*Azospirillum brasilense*	Nitrogen fixation	Maize (*Zea mays*)	[7]
*Bradyrhizobium* sp.	Nitrogen fixation	Soybean (*Glycine max*)	[8]
*Pseudomonas* sp.	Phosphate solubilization	Pea (*Pisum sativum*)	[9]
*Pseudomonas* sp.	Phosphate solubilization	Maize (*Z. mays*)	[10]
*Pseudomonas* *brassicae*	Siderophore production	Mung bean (*Vigna radiata*)	[11]
*Pseudomonas aeruginosa*	Phytohormone production	Mung bean (*V. radiata*)	[12]
*Pseudomonas putida*	Antioxidant activity	Maize (*Z. mays*)	[13]
*Bacillus velezensis*	Phosphate solubilization and phytohormone production	Wheat (*Triticum aestivum*)	[14]
*B. velezensis*	Phosphate solubilization	Soybean (*G. max*) and Maize (*Z. mays*)	[15]
*Bacillus mojavensis*	Volatile organic compounds (VOCs)’ production	*Arabidopsis thaliana*	[16]
*Bacillus subtilis*	1-aminocyclopropane-1-carboxylic acid (ACC) deaminase activity	Tomato (*Solanum lycopersicum*)	[17]
*Serratia* sp.	ACC deaminase activity and phytohormone production	Sunflower (*Helianthus annuus*)	[18]

**Table 2 plants-13-02246-t002:** Some examples of PGPF growth promotion mechanisms.

Specie	Mechanism of Growth Promotion	Culture	Reference
*Trichoderma* sp.	Phosphate solubilization	Soybean (*Glycine max*)	[57]
*Trichoderma* sp.	VOCs production	*Arabidopsis thaliana*	[58]
*Trichoderma koningiopsis*	VOCs production	*A. thaliana*	[59]
*Trichoderma viride*	VOCs production	Tomato (*Solanum lycopersicum*)	[60]
*Trichoderma asperellum*	Phosphate solubilization and phytohormone production	Cucumber (*Cucumber sativus*)	[61]
*Rhizophagus clarus*	Increase in P and N content	Soybean (*G. max*) and Cotton (*Gossypium hirsutum*)	[62]
*Glomus intraradices*	Salt tolerance	Wheat (*Triticum aestivum*)	[63]
*Rhizophagus irregularis*	Drought tolerance	Maize (*Zea mays*)	[64]
Mix of *R. clarus*, *R. intraradices*, *Septoglomus deserticola*, *Funneliformis mosseae*	Water deficit tolerance	Soybean (*G. max*)	[65]
*Rhizophagus irregularis*, *F. mosseae*, and *Funneliformis geosporum*	High temperature tolerance	Soybean (*G. max*)	[66]

**Table 3 plants-13-02246-t003:** Benefits of inoculating microorganisms.

Species	Culture	Benefits	References
*Bradyrhizobium diazoefficiens* and *Rhizobium tropici*	Common beans (*Phaseolus vulgaris*)	Growth promotion and grain yield	[108]
*Bradyrhizobium japonicum* and *Azospirillum brasilense*	Soybean (*Glycine max*)	Increased yield components, grain yield, and seed quality	[109]
*Pseudomonas fluorescens* and *A. brasilense*	Tomato (*Solanum lycopersicum*)	Increased yield and fruit quality	[110]
*Bradyrhizobium* sp. and *Trichoderma* sp.	Cowpea (*Vigna unguiculata*)	Increased the growth rate, biomass, and photosynthetic pigments	[111]
*Bacillus licheniformis* and *Bacillus subtilis*	Cucumber (*Cucumber sativus*)	Alleviated salt stress	[112]
*B. subitilis*, *Bacillus megaterium* and *Rhizophagus intraradices*	Soybean (*G. max*)	Increase leaf nutrient and in yield	[113]
*Rhizophagus irregulares* and *Bradyrhizobium* sp.	Mung bean (*Vigna radiata*)	Growth promotion and alleviated water stress	[106]

**Table 4 plants-13-02246-t004:** Genus of microorganisms registered for the main crops in Brazil, organized in single inoculation or co-inoculation.

Culture	Genus	Single Inoculation	Co-Inoculation
Soybean (*Glycine max*)	*Bradyrhizobium* sp.	246	*Bradyrhizobium Japonicum* + *Bradyrhizobium Elkani* (52) *B. Japonicum* + *Azospirillum brasilense* (4) *A. brasilense* + *Pseudomonas fluorescense* (2) *Bacillus megaterium + Bacillus subtilis* (2) *B. subtilis* + *B. elkani* (3) *Rhizoglomus intraradices* + *Claroideoglomus claroideum* (2) *B. subtilis* + *B. elkani +Parabhurkodelia nodosa* (3) *B. subtilis* + *Bacillus amyloliquefacens +Bacillus pumilus* (1) *P. fluorescense* + *B. amyloliquefacens + Priestia megaterium* (1)
	*Azospirillum* sp.	28	
	*Bacillus* sp.	8	
	*Pseudomonas* sp.	6	
	*Trichoderma* sp.	5	
	*Rhizophagus* sp. (*Rhizoglomus*)	5	
Common beans (*Phaseolus vulgaris*)	*Rhizobium* sp.	65	*B. megaterium* + *B. subtillis* (1)
	*Azospirilum* sp.	6	
	*Bacillus* sp.	1	
Maize (*Zea mays*)	*Azospirillum* sp.	41	*B. megaterium* + *B. subtilis* (2) *A. brasilense* + *P. fluorescense* (2) *B. megaterium* + *Lysinobacillus* sp. (1) *B. licheniformis* + *Bacillus aryabhattai* (2) *B. Japonicum* + *A. brasilense* (2) *R. intraradices* + *C. claroideum* (2) *B. subtilis* + *B. amyloliquefacens* +*B. pumilus* (1)
	*Bacillus* sp.	8	
	*Rhizophagus* sp. (*Rhizoglomus*)	5	
	*Pseudomonas* sp.	3	
	*Methylobacterium* sp.	1	
Peanut (*Arachis hypogaea*)	*Bradyrhizobium* sp.	36	-
Jack bean (*Canavalia ensiformis*)	*Bradyrhizobium* sp.	35	-
Wheat (*Triticum aestivum*)	*Azospirilum* sp.	23	-

## Data Availability

Data are contained within the article.

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
