# Peer review of "Microbial Fertilizers: A Study on the Current Scenario of Brazilian Inoculants and Future Perspectives"

_plants, 2024, doi:10.3390/plants13162246_

Round 1

Reviewer 1 Report

Comments and Suggestions for Authors

Your paper provides a comprehensive and detailed analysis of the current trends, challenges, and future potential of biological inoculants in Brazilian agriculture. Here are some holistic comments and suggestions to enhance the clarity, readability, and impact of your paper:

Ensure that the introduction clearly outlines the scope and objectives of the paper. Briefly introduce the importance of biological inoculants in modern agriculture and why this topic is relevant for Brazil. Define all acronyms (e.g., NFB, AMF, PSB) at their first mention to ensure clarity for all readers.

Arrange sections in a logical flow, moving from general information to specific details. Consider using subheadings to break down complex sections into more digestible parts. Break down long paragraphs into shorter, more focused sections to improve readability and help maintain the reader's attention.

While the detailed information is valuable, ensure a balance between depth and readability. Avoid overwhelming the reader with too much technical detail in one section. Utilize tables, figures, and bullet points to present data and statistics clearly, helping readers quickly grasp key information and trends.

In the section on Brazil's biological product market, provide more context on the economic and environmental impact of current practices. Highlight the importance of reducing dependency on imported fertilizers and the potential benefits of biological inoculants. Clarify the issues surrounding phosphorus and potassium fertilization, including the inefficiencies and economic costs. Discuss how AMF and PSB can address these issues and the current gaps in their adoption. Expand on the benefits and challenges of co-inoculation, providing specific examples of successful practices and their outcomes. Elaborate on the types of stress mitigators currently available and their potential impact on crop resilience and yield, discussing the need for further research and development in this area.

The conclusion should succinctly summarize the key findings and insights from the paper, emphasizing the potential of biological inoculants to enhance sustainability and reduce costs in Brazilian agriculture. Offer a clear vision for the future of biological inoculants in Brazil, highlighting areas where further research and investment are needed and the potential long-term benefits for the agricultural sector.

Ensure consistency in terminology, acronyms, and formatting throughout the paper. Aim for clear and concise language, avoiding jargon where possible, or providing explanations when technical terms are used. Ensure all references are up-to-date and relevant, citing sources accurately to support your claims and provide a robust academic foundation for your arguments.

Your paper addresses an important and timely topic in agricultural science. By incorporating these suggestions, you can enhance the clarity, impact, and overall quality of your work. Thank you for your valuable contribution to the field.

These comments and suggestions aim to provide a constructive critique to help you refine and improve your paper. Your thorough research and analysis are commendable, and with some adjustments, your paper can significantly contribute to the understanding and advancement of biological inoculants in Brazilian agriculture.

Comments on the Quality of English Language

The manuscript generally demonstrates a good command of English language. The writing is clear and mostly coherent, effectively conveying complex scientific concepts. However, there are instances where sentence structures could be further refined for better readability and flow. Some paragraphs appear dense and could benefit from simplification or restructuring to improve clarity and coherence. Additionally, attention to grammatical accuracy, particularly in terms of verb tense consistency and sentence construction, would enhance the overall polish of the manuscript. Overall, enhancing the clarity and precision of language will strengthen the manuscript's impact and accessibility to a wider scientific audience.

Author Response

Reviewer 1

Your paper provides a comprehensive and detailed analysis of the current trends, challenges, and future potential of biological inoculants in Brazilian agriculture. Here are some holistic comments and suggestions to enhance the clarity, readability, and impact of your paper:

Ensure that the introduction clearly outlines the scope and objectives of the paper. Briefly introduce the importance of biological inoculants in modern agriculture and why this topic is relevant for Brazil.

A paragraph has been added to the introduction highlighting the importance of agriculture to the country and the role of inoculants in promoting more sustainable agricultural practices.

Define all acronyms (e.g., NFB, AMF, PSB) at their first mention to ensure clarity for all readers.   

The acronyms in the text have been reviewed.

Arrange sections in a logical flow, moving from general information to specific details. Consider using subheadings to break down complex sections into more digestible parts. Break down long paragraphs into shorter, more focused sections to improve readability and help maintain the reader's attention. While the detailed information is valuable, ensure a balance between depth and readability. Avoid overwhelming the reader with too much technical detail in one section. Utilize tables, figures, and bullet points to present data and statistics clearly, helping readers quickly grasp key information and trends.

Some information has been removed from the text in the section on Brazil's biological product market (Topic 5), related to Figure 2 and Table 4, in order to reduce the amount of data presented. Additionally, this section has been divided into subsections to facilitate the flow of reading and understanding of the text:

5.1. Formulations: single and coinoculation

5.2. Biofertilization

5.2. Abiotic stress mitigation

In the section on Brazil's biological product market, provide more context on the economic and environmental impact of current practices. Highlight the importance of reducing dependency on imported fertilizers and the potential benefits of biological inoculants. Clarify the issues surrounding phosphorus and potassium fertilization, including the inefficiencies and economic costs. Discuss how AMF and PSB can address these issues and the current gaps in their adoption. Expand on the benefits and challenges of co-inoculation, providing specific examples of successful practices and their outcomes. Elaborate on the types of stress mitigators currently available and their potential impact on crop resilience and yield, discussing the need for further research and development in this area.

In subsection 5.2, Biofertilization, a paragraph (the first) has been added to contextualize Brazil's economic dependence on the importation of chemical fertilizers and to explain why these inputs are poorly utilized and have low efficiency. Additionally, information has been included in paragraphs 3, 4, 6, and 7 to enhance the understanding of the importance of AMF, PSB, and potassium-solubilizing bacteria in reducing the application of these fertilizers.

The conclusion should succinctly summarize the key findings and insights from the paper, emphasizing the potential of biological inoculants to enhance sustainability and reduce costs in Brazilian agriculture. Offer a clear vision for the future of biological inoculants in Brazil, highlighting areas where further research and investment are needed and the potential long-term benefits for the agricultural sector.

The topic "Future Perspectives" has been added, discussing the future prospects of the use and research of inoculants in Brazil. Additionally, the conclusion has been revised.

Ensure consistency in terminology, acronyms, and formatting throughout the paper. Aim for clear and concise language, avoiding jargon where possible, or providing explanations when technical terms are used. Ensure all references are up-to-date and relevant, citing sources accurately to support your claims and provide a robust academic foundation for your arguments.

The text has been reviewed and revised once again for language accuracy.

Your paper addresses an important and timely topic in agricultural science. By incorporating these suggestions, you can enhance the clarity, impact, and overall quality of your work. Thank you for your valuable contribution to the field.

These comments and suggestions aim to provide a constructive critique to help you refine and improve your paper. Your thorough research and analysis are commendable, and with some adjustments, your paper can significantly contribute to the understanding and advancement of biological inoculants in Brazilian agriculture.

Thank you for your feedback!

Reviewer 2

This review deal with microbial fertilizers by inoculants as future perspective. Is an interesting work focused in description of beneficial effects of  However, currently we applied this inoculant on crops. Even, you reported in introduction section huge amount of these microorganisms. Under this conext do you consider as future perspective?.

Although these microorganisms are already being applied today, this market remains limited in terms of mechanisms; a large portion of the applied inoculants are nitrogen-fixing bacteria. The article aims to provide a future perspective on the microorganisms and mechanisms that could be registered in accordance with the challenges faced by Brazilian agriculture. Within this context, we agree that the use of inoculants is already a part of Brazilian agricultural systems, but we understand that there is significant room for expansion.

What is the role of soil on inoculant persistence? How is their association?

Although it is a relevant discussion, the authors believe it does not fit within the scope of the article. The idea was to review the main mechanisms of action of PGPR and PGPF as a basis for discussing the scenario of registered inoculants in the Brazilian market. The inclusion of other aspects related to interactions could detract from the main theme. 

Figure 1: Please add references about mechanisms. This figure (as graphical abstract) could represent the topics addressed in the review. 

Mechanisms that were not adequately described in the text have been removed from the figure.

Table 1: Bacillus velezensis, in italic

TAble 2: Water déficit tolerance, correct

Corrections have been made regarding Table 1 and Table 2.

Review reports interesting and supported information, but some sections show weakness in comparison to other, i.e. section 2.3 and 2.5.  Please reinforce this topics. 

Topics 2.3 and 2.5 have been reformulated, with an expanded explanation of the mechanisms of action of the main phytohormones produced by PGPR (2.3). In Topic 2.5, examples have been added to highlight the importance of the production of volatile compounds.

Topic 2.2 What happen with non-enzymatic antioxidant compounds?

Examples of non-enzymatic antioxidant compounds have been added to the topic.

Conclusion did not mention clearly  the perspectives. 

The topic "Future Perspectives" has been added, discussing the future prospects for the use and research of inoculants in Brazil. Additionally, the conclusion has been revised.

Finally, I suggest revision of english language 

I suggest following work

Acuña, J. J., Jorquera, M. A., Martínez, O. A., Menezes-Blackburn, D., Fernández, M. T., Marschner, P., ... & Mora, M. L. (2011). Indole acetic acid and phytase activity produced by rhizosphere bacilli as affected by pH and metals. Journal of soil science and plant nutrition11(3), 1-12.

The text has been reviewed and revised once again for language accuracy. 

Thank you for your feedback!

Reviewer 3

I commend the authors for topic addressed and the experimental concept.

The subject addressed is of great interest in the context of the new legislative framework regarding fertilizing products and modern technologies in the agricultural field, in the context of major climate changes and the principles of a circular economy.

Considering the socio-economic importance of the field, the studies should be directed, financed and carried out in the future and validated in experimental field conditions and different climatic conditions. However, special attention and new studies should be given regarding the possible phytotoxic effects and on food safety and security through the application of microbial agents in agriculture, limiting their inclusion in the existing, legislated and approved categories of fertilizing products.

Reviewer 2 Report

Comments and Suggestions for Authors

Dear authors 

This review deal with microbial fertilizers by inoculants as future perspective. Is an interesting work focused in description of beneficial effects of  However, currently we applied this inoculant on crops. Even, you reported in introduction section huge amount of these microorganisms. Under this conext do you consider as future perspective?.

What is the role of soil on inoculant persistence? How is their association?

Figure 1: Please add references about mechanisms. This figure (as graphical abstract) could represent the topics addressed in the review. 

Table 1: Bacillus velezensis, in italic

TAble 2: Water déficit tolerance, correct

Review reports interesting and supported information, but some sections show weakness in comparison to other, i.e. section 2.3 and 2.5.  Please reinforce this topics. 

Topic 2.2 What happen with non-enzymatic antioxidant compounds?

Conclusion did not mention clearly  the perspectives. 

Finally, I suggest revision of english language 

I suggest following work

Acuña, J. J., Jorquera, M. A., Martínez, O. A., Menezes-Blackburn, D., Fernández, M. T., Marschner, P., ... & Mora, M. L. (2011). Indole acetic acid and phytase activity produced by rhizosphere bacilli as affected by pH and metals. Journal of soil science and plant nutrition11(3), 1-12.

Comments on the Quality of English Language

Manuscript needs check english language by native english speaker.

Author Response

This review deal with microbial fertilizers by inoculants as future perspective. Is an interesting work focused in description of beneficial effects of  However, currently we applied this inoculant on crops. Even, you reported in introduction section huge amount of these microorganisms. Under this conext do you consider as future perspective?.

Although these microorganisms are already being applied today, this market remains limited in terms of mechanisms; a large portion of the applied inoculants are nitrogen-fixing bacteria. The article aims to provide a future perspective on the microorganisms and mechanisms that could be registered in accordance with the challenges faced by Brazilian agriculture. Within this context, we agree that the use of inoculants is already a part of Brazilian agricultural systems, but we understand that there is significant room for expansion.

What is the role of soil on inoculant persistence? How is their association?

Although it is a relevant discussion, the authors believe it does not fit within the scope of the article. The idea was to review the main mechanisms of action of PGPR and PGPF as a basis for discussing the scenario of registered inoculants in the Brazilian market. The inclusion of other aspects related to interactions could detract from the main theme. 

Figure 1: Please add references about mechanisms. This figure (as graphical abstract) could represent the topics addressed in the review. 

Mechanisms that were not adequately described in the text have been removed from the figure.

Table 1: Bacillus velezensis, in italic

TAble 2: Water déficit tolerance, correct

Corrections have been made regarding Table 1 and Table 2.

Review reports interesting and supported information, but some sections show weakness in comparison to other, i.e. section 2.3 and 2.5.  Please reinforce this topics. 

Topics 2.3 and 2.5 have been reformulated, with an expanded explanation of the mechanisms of action of the main phytohormones produced by PGPR (2.3). In Topic 2.5, examples have been added to highlight the importance of the production of volatile compounds.

Topic 2.2 What happen with non-enzymatic antioxidant compounds?

Examples of non-enzymatic antioxidant compounds have been added to the topic.

Conclusion did not mention clearly  the perspectives. 

The topic "Future Perspectives" has been added, discussing the future prospects for the use and research of inoculants in Brazil. Additionally, the conclusion has been revised.

Finally, I suggest revision of english language 

I suggest following work

Acuña, J. J., Jorquera, M. A., Martínez, O. A., Menezes-Blackburn, D., Fernández, M. T., Marschner, P., ... & Mora, M. L. (2011). Indole acetic acid and phytase activity produced by rhizosphere bacilli as affected by pH and metals. Journal of soil science and plant nutrition11(3), 1-12.

The text has been reviewed and revised once again for language accuracy. 

Thank you for your feedback!

Reviewer 3 Report

Comments and Suggestions for Authors

I commend the authors for topic addressed and the experimental concept.

The subject addressed is of great interest in the context of the new legislative framework regarding fertilizing products and modern technologies in the agricultural field, in the context of major climate changes and the principles of a circular economy.

Considering the socio-economic importance of the field, the studies should be directed, financed and carried out in the future and validated in experimental field conditions and different climatic conditions. However, special attention and new studies should be given regarding the possible phytotoxic effects and on food safety and security through the application of microbial agents in agriculture, limiting their inclusion in the existing, legislated and approved categories of fertilizing products.

Author Response

I commend the authors for topic addressed and the experimental concept.

The subject addressed is of great interest in the context of the new legislative framework regarding fertilizing products and modern technologies in the agricultural field, in the context of major climate changes and the principles of a circular economy.

Considering the socio-economic importance of the field, the studies should be directed, financed and carried out in the future and validated in experimental field conditions and different climatic conditions. However, special attention and new studies should be given regarding the possible phytotoxic effects and on food safety and security through the application of microbial agents in agriculture, limiting their inclusion in the existing, legislated and approved categories of fertilizing products.
